# Standard Radio-Iodine Labeling Protocols Impaired the Functional Integrity of Mesenchymal Stem/Stromal Cell Exosomes

**DOI:** 10.3390/ijms25073742

**Published:** 2024-03-27

**Authors:** Chang-Tong Yang, Ruenn Chai Lai, Vanessa Jing Xin Phua, Swee Eng Aw, Bin Zhang, Wei Kian Sim, Sai Kiang Lim, David Chee Eng Ng

**Affiliations:** 1Department of Nuclear Medicine and Molecular Imaging, Radiological Sciences Division, Singapore General Hospital, Outram Road, Singapore 169608, Singapore; vanessa.phua.x.j@sgh.com.sg (V.J.X.P.); david.ng.c.e@singhealth.com.sg (D.C.E.N.); 2Duke-NUS Medical School, 8 College Road, Singapore 169857, Singapore; 3Institute of Molecular and Cell Biology (IMCB), Agency for Science, Technology and Research (A*STAR), 61 Biopolis Drive, Proteos, Singapore 138673, Singapore; lai_ruenn_chai@imcb.a-star.edu.sg (R.C.L.); zhang_bin@imcb.a-star.edu.sg (B.Z.); eugene_sim@imcb.a-star.edu.sg (W.K.S.); 4Paracrine Therapeutics Pte. Ltd., 10 Choa Chu Kang Grove #13-22 Sol Acres, Singapore 688207, Singapore

**Keywords:** radiolabeling, iodination, mesenchymal stem/stromal cell (MSC), exosome, CD73 enzymatic activity

## Abstract

Mesenchymal stem/stromal cells (MSCs) are an extensively studied cell type in clinical trials due to their easy availability, substantial ex vivo proliferative capacity, and therapeutic efficacy in numerous pre-clinical animal models of disease. The prevailing understanding suggests that their therapeutic impact is mediated by the secretion of exosomes. Notably, MSC exosomes present several advantages over MSCs as therapeutic agents, due to their non-living nature and smaller size. However, despite their promising therapeutic potential, the clinical translation of MSC exosomes is hindered by an incomplete understanding of their biodistribution after administration. A primary obstacle to this lies in the lack of robust labels that are highly sensitive, capable of directly and easily tagging exosomes with minimal non-specific labeling artifacts, and sensitive traceability with minimal background noise. One potential candidate to address this issue is radioactive iodine. Protocols for iodinating exosomes and tracking radioactive iodine in live imaging are well-established, and their application in determining the biodistribution of exosomes has been reported. Nevertheless, the effects of iodination on the structural or functional activities of exosomes have never been thoroughly examined. In this study, we investigate these effects and report that these iodination methods abrogate CD73 enzymatic activity on MSC exosomes. Consequently, the biodistribution of iodinated exosomes may reflect the biodistribution of denatured exosomes rather than functionally intact ones.

## 1. Introduction

Mesenchymal stromal/stem cells (MSCs) are an adult cell type that has been extensively investigated in clinical trials across various disease indications. MSCs are minimally defined by the International Society for Cellular Therapy (ISCT) as being plastic-adherent when maintained in standard culture conditions, expressing CD105, CD73, and CD90, and lacking expression of CD45, CD34, CD14 or CD11b, CD79a, or CD19 and HLA-DR surface molecules, and must differentiate osteoblasts, adipocytes, and chondroblasts in vitro [1]. They could be isolated from diverse sources, including bone marrow, adipose tissue, and umbilical cord blood [2]. Unlike the initial hypothesis suggesting that transplanted MSCs exert therapeutic effects by homing to injured tissues, engrafting, and differentiating into appropriate cell types to replace damaged tissues, it is now widely accepted that MSCs achieve their therapeutic activity through secreted factors rather than direct differentiation [3,4,5]. In 2008, it was reported that the active secreted factors had a molecular weight exceeding 1000 Kd [6]. Subsequently, in 2009 and 2010, MSCs were first shown to exert their therapeutic effects through secreted extracellular vesicles (EVs), namely microvesicles ranging from 80 to 1000 nm [7] and MSC exosomes with a size range of 100–130 nm [8]. Notably, in 2017, Cammussi and colleagues revealed that among the 80–1000 nm microvesicles, it was the smaller ~160 nm fraction, rather than the larger ~215 nm fraction, that exhibited therapeutic properties [9].

Importantly, MSC-sEVs demonstrated therapeutic efficacy comparable to that of their parental MSCs [7,10,11,12]. The widely recognized therapeutic activity of MSCs against various diseases is now mainly attributed to small EVs or sEVs in the 50–200 nm size range [13,14,15,16,17]. In a direct head-to-head comparison between the therapeutic activity of MSCs and their EV products, MSC EVs could recapitulate the therapeutic activity of MSCs [7,10,11]. The initially reported 100–130 nm small EV preparations by Lai et al. [8] were later determined to contain bona fide exosomes derived from endosomes [18], qualifying them as exosome preparations. Compared with MSC cell-based therapy, exosome-based therapy offers tremendous advantages over cell-based therapy. It is smaller in size and non-living; therefore, it is safer and easier to store, transport, and administer [19].

However, the clinical translation of MSC exosomes faces challenges due to a lack of effective methods to label and track the biodistribution of exosomes or small EVs. Establishing a spatiotemporal biodistribution congruent with a proposed model is essential for elucidating their mechanism of action, as previously outlined [20,21,22]. Radiolabeling currently stands out as the most sensitive and effective method for directly labeling and tracking EVs in vivo, allowing for deep tissue penetration in biodistribution studies [23,24].

Radioactive iodine, specifically ^131^iodine [25,26] and ^124^iodine [27], has been used to study the biodistribution of EVs. The advantage of using radioactive iodine lies in the ease of attaching these labels covalently and directly to membranes without genetic manipulation of the EV-producing cells or the need to use an intermediary chelator. Also, unlike fluorescent dyes, radioactive iodine labels have low background and provide more sensitive live imaging of their distribution. Another advantageous feature of, e.g., ^131^iodine, is its relatively long half-life of 8 days, making it well-suited for long-tracking kinetics. The method for iodine labeling is well-established, further enhancing its suitability for tracking purposes.

We investigated the effects of iodination using both the chloramine-T and Pierce iodination methods on the functional integrity of MSC exosomes produced by a monoclonal immortalized E1 mesenchymal stem cell line as described in the method by monitoring the enzymatic activity of CD73, a surface antigen on MSC exosomes. The use of monoclonal immortalized MSCs reduces variability in MSC exosomes caused by differences in EV-producing cells. Chloramine-T and Pierce iodination methods have been employed for exosome labeling. Despite their somewhat antiquated nature, these methods offer distinct advantages in addressing challenges associated with exosome labeling and tracking, both in vitro and in vivo. Firstly, the extended half-life of ^131^I- (8 days) facilitates the desired tracking kinetics, and if PET imaging is preferred, the ^124^I isotope (4.2 days half-life) with 25.6% positron emission can be employed. Secondly, the well-established chemistry of iodine radiolabeling to peptides and antibodies ensures optimal radiolabeling yields and high specific labeling activity, a crucial consideration given the relatively low exosome dosing (1–10 µg per mouse or 40–400 µg per kg) compared to antibody dosing, e.g., 10 mg/kg bevacizumab dosing in mice [28]. Finally, the straightforward elimination of unlabeled free iodine after labeling ensures the high purity of the labeled exosomes, in stark contrast to lipophilic fluorescent dyes. These fluorescent dyes spontaneously form micelles in aqueous solutions, and these micelles cannot be separated from the labeled exosomes, resulting in nonspecific labeling artifacts [29,30,31].

Despite several reports on the use of radioactive iodine-labeling of EVs for biodistribution studies, the effects of iodination on the structural or functional integrity of EVs were never investigated. Here, we report that both methods significantly impaired the functional activity of the MSC exosomes as demonstrated by the abrogated enzymatic activity of CD73, a surface protein on MSC exosomes. This impaired functionality is likely the result of denaturation caused by the iodination process. Denatured exosomes are susceptible to similar metabolic breakdown as other denatured biological complexes or cells. Consequently, their biodistribution is expected to reflect this process more than that of pharmacologically active intact exosomes.

## 2. Results

### 2.1. Radiolabeling

The radiolabeling yields of exosomes labeled using chloramine-T and Pierce iodination methods were 35% and 10%, respectively, based on the iTLC chromatogram in Figure 1A,C. After purification through a PD-10 column, the purity of ^131^I-labeled exosomes exceeded 90% in Figure 1B,D.

The ^131^I-labeled exosomes from both labeling methods were incubated at 37 °C in PBS containing 20% fetal bovine serum (FBS) for 1, 4, and 24 h, and monitored for loss of radioactivity. The stability of the ^131^I-labeled exosomes was monitored for the release of ^131^I using an iTLC chromatogram. The amount of ^131^I released from the labeled exosomes was negligible at 1 and 4 h. The total amount of ^131^I fraction released from ^131^I-labeled exosomes was less than 10% of overall activity during the 24 h incubation period, indicating the ^131^I-labeled exosomes are stable in FBS.

### 2.2. Integrity of Exosomes after Iodine Labeling

To assess the integrity of exosomes before and after iodine labeling, the exosomes were labeled with cold iodine using the two labeling methods, namely chloramine T and Pierce iodination labeling protocols. The size of the particles was assessed by Nanoparticle Tracking Analysis (NTA)(Table 1). Before labeling, the size of the particles was 131.25 ± 15.91 nm. After labeling, they were 148.39 ± 11.85 nm and 132.57 ± 27.9 nm. The differences in sizes before and after labeling were not statistically significant.

To determine the functional integrity of the exosomes after iodine labeling, the CD73 enzymatic activity was determined before and after labeling with cold NaI using chloramine T and Pierce labeling protocols (Table 1). CD73 is a defining feature of MSC exosomes [13]. CD73 is a surface ecto-5′-nucleotidase (NT5E) that catalyzes the hydrolysis of nucleotide monophosphates. Incubation with the cold NaI alone has no effect on CD73 enzyme activity. However, labeling using chloramine T or Pierce iodination reduced CD73 activity to 9.7 ± 3.1% (Figure 2) and 66.6 ± 1.9% of the initial activity (Figure 2), respectively.

To assess if the functional integrity of exosomes is affected by exposure to high levels of radiation, the exosomes were exposed to ~12 mCi (4× amount of radiation strength used for exosomes radiolabeling) at −20 °C for 1, 4, and 7 days. There was no statistically significant difference in CD73 enzymatic activity at all time points compared with no treatment (Figure 3).

## 3. Discussion

MSC exosomes have exhibited therapeutic efficacy in numerous preclinical animal disease models. However, a significant challenge in advancing MSC exosomes as therapeutic drugs lies in understanding the mechanism underlying their therapeutic effectiveness. A prerequisite for a proposed mechanism of action is establishing their compatibility with the spatiotemporal distribution of exosomes.

However, as discussed in the introduction, labeling exosomes for biodistribution studies has been a major challenge in exosome research. Here, we consider radiolabeling exosomes with iodine because (1) iodine labeling is a well-established method; (2) its isotope ^131^iodine has a relatively long half-life (8 days), which is suitable for long-tracking kinetics, and ^125^iodine can label exosomes for PET imaging; and (3) iodine can be labeled to the proteins in exosomes covalently or directly to membranes without genetic manipulation of the exosome-producing cells. Although radioiodination has several attractive features, and radioiodination of exosomes to study biodistribution has been reported, our study demonstrates that the standard methods of iodination (namely the chloramine-T and the Pierce iodination methods) may not be suitable for labeling exosomes or EVs, as they destroy the functional integrity of the exosomes, possibly through oxidative denaturation as evidenced by the smaller loss in enzymatic activity with the exosomes iodinated with the milder oxidizing Pierce iodination method [32]. As denatured exosomes would be metabolically broken down as per other denatured biological complexes or cells, their biodistribution will be expected to reflect this metabolic breakdown process more than that of pharmacologically active intact exosomes. Hence, biodistribution studies of radio-iodinated exosomes labeled using the chloramine T and the Pierce iodination methods will not reveal the biodistribution of pharmacologically active exosomes.

Our study demonstrated that radioiodination of MSC exosomes for biodistribution studies will require optimization of the current protocols. A key consideration is to avoid the harsh oxidative conditions of iodination. One way to circumvent this condition is to use an indirect method of labeling, such as the use of a streptavidin–lactadherin fusion protein and iodine-125-labeled biotin derivative [33]. However, this method may not be universally applicable to all EVs, as not all EVs, including exosomes, bind lactadherin.

## 4. Materials and Methods

Chloramine-T and sodium metabisulfite Na_2_S_2_O_5_ were purchased from Sigma-Aldrich, Singapore and used as received. Pierce^TM^ iodination reagent (1,3,4,6-tetrachloro-3α,6α-diphenyl-glycoluril), Pierce^TM^ iodination beads, and tubes were purchased from Thermo Scientific. Phosphate buffered saline (PBS, pH 7.4) was purchased from Sigma-Aldrich. PD-10 desalting column (sephadex^TM^ G-25M) was purchased from GE Healthcare Bio-Science AB, Sweden. The PD-10 column is conditioned by filling the column with equilibration buffer, allowing the equilibration buffer to enter the packed bed completely, and then discarding the flow-through. This was repeated four times. A Bio-Spin^®^ column with Bio-Gel^®^ P-30 was purchased from Bio-Rad, Hercules, CA, USA, prehydrated in Tris buffer, and used with no need to be conditioned. ^131^I in Na^131^I solution was purchased from the Radioisotope Centre POLATOM, Poland. The radioactivity measurement of ^131^I was performed using an AtomLabTM 500 dose calibrator from Biodex Medical Systems, Inc., New York, NY, USA. Thin-layer chromatography (TLC) silica gel 60 F254 aluminum sheets (20 × 20 cm) were purchased from Merck KGaA, Germany and cut into 2 × 10 cm strips. Instant thin-layer chromatography (iTLC) was performed using a Bioscan AR-2000, Wilmington, MA, USA, with a P10 cylinder containing methane and argon gas. All radioactive substances were handled according to the guidelines of the Radiation Protection and Nuclear Sciences Department, National Environment Agency (NEA). All the experiments were performed according to NEA guidelines and regulations. MSC–exosomes were provided by the Institute of Molecular and Cell Biology, which were prepared from immortalized E1-MYC 16.3 human ESC-derived mesenchymal stem cells [34].

### 4.1. Preparation and Culture Conditions of MSC Exosomes

The exosome used in the manuscript was generated from a single MSC clone, which is a monoclonal immortalized E1-MYC 16.3 human ESC-derived mesenchymal stem cell line [34] which was cultured in DMEM with 10% fetal calf serum. For MSC–exosome preparation, the conditioned medium was prepared by growing 80% confluent cells in a chemically defined medium for three days, as previously described [8]. The defined medium was prepared as follows: 480 mL Dulbecco’s modified eagle medium (DMEM, 31053, Thermo Fisher, Waltham, MA, USA), 5 mL Non-essential amino acids (NEAA, 11140–050, Thermo Fisher, Waltham, MA, USA), 5 mL L Glutamine (25030–081, Thermo Fisher, Waltham, MA, USA), 5 mL Sodium Pyruvate (11360, Thermo Fisher, Waltham, MA, USA), 5 mL ITS–X (51500–056, Thermo Fisher, Waltham, MA, USA), and 0.5 mL 2–ME (21985−02, Thermo Fisher, Waltham, MA, USA). This was supplemented with 0.1 mL basic fibroblast growth factor (bFGF, 0.5 ng/μL 0.2% BSA in PBS (+)) and 0.005 mL Platelet-derived (PDGF, 100 ng/μL PBS (+)). These latter components were obtained as follows: Bovine Serum Albumin or BSA (A9647, Sigma–Aldrich, St. Louis, MO, USA), PDGF (100–00 AB CYTOLAB), bFGF (13256–029, Thermo Fisher, Waltham, MA, USA) and PBS (+) (14040–133, Thermo Fisher, Waltham, MA, USA). The conditioned medium (CM) was size–fractionated by tangential flow filtration and then concentrated 50× using a membrane with a molecular weight cut–off (MWCO) of 100 kDa (Sartorius, Gottingen, Germany). The MSC–exosome preparations were characterized as per the MSC-sEV defining identity proposed by a consensus of ISCT, ISEV, ISBT, and SOCRATES [13]. Protein concentration was assayed using a Coomassie Plus (Bradford) Assay Kit (Thermo Fisher Scientific, Waltham, MA, USA), and cholesterol concentration was assayed using a Cholesterol Quantitation Kit (Abcam, Cambridge, UK). The expression of CD81 and CD73 in the preparations was confirmed by Western or ELISA [8,34,35]. The MSC–exosome preparations were filtered with a 0.22 μm filter (Merck Millipore, Billerica, MA, USA) and stored in a −80 °C freezer.

### 4.2. Radiolabeling

Both chloramine T [36,37,38] and Pierce iodination [39,40,41] methods with little modification have been employed for radioiodination labeling of exosomes. Exosomes are labeled with ^131^I using the chloramine T method, and 50 μg of exosomes dissolved in 0.1 mL PBS (pH 7.4, 1×) in a 2 mL microtube (Axygen scientific^®^, Union City, CA, USA). Add 0.1–0.15 mL of the Na^131^I solution (about 2–3 mCi) to the above exosomes in the tube. Then, 0.1 mL of the above fresh prepared chloramine-T solution (4 mg/mL) was added and mixed for at least 1 min. The reaction time can be extended to 2–3 min for labeling. Then, 0.1 μL of Na_2_S_2_O_5_ solution (2.4 mg/1 mL) was added and stirred for 30 s to terminate the iodination reaction. Purify the iodinated exosomes from excess reactants by running the PD-10 column. The column was pre-treated by passing a solution of PBS through it to eliminate nonspecific binding sites that could cause exosome loss. Excess free iodine is separated and removed from the labeled MSC exosomes using the size exclusion PD-10 column. Once the entire MSC exosome-radioiodine reaction mixture entered into the column matrix, PBS was added to fill the column reservoir and collect twenty 500 μL fractions in test tubes. The radioactivity in each fraction was measured and the first peak fraction collected, which represents radioiodinated MSC exosomes while the subsequent flat peak represents free iodine. Exosomes labeled with non-radioactive iodine will be used to determine if iodine incorporation compromises the structural and biochemical integrity of MSC exosomes. The stability of iodine-labeled exosomes in serum will also be assessed.

The previously reported Pierce iodination method has been used and modified for radioiodination labeling [42,43,44]. Around 3 mCi of ^131^I solution was added directly to the Iodogen-coated tube or four Iodogen beads in the test tube containing 0.1 mL of PBS. Then, the tube was gently swirled at room temperature for 5 min. The 50 μg of MSC exosomes resuspended in 0.1 mL PBS were added to the reaction tube. The mixture was incubated at room temperature for 30 min and then purified through the PD-10 column to separate the iodinated MSC exosomes from free iodine using the same purification method as described above for the chloramine T procedures.

### 4.3. Nanoparticle Tracking Analysis

Particle measurement of MSC exosomes was determined by Nanoparticle tracking analysis on a ZetaView instrument (Particle Matrix GmbH, Ammersee, Germany) using the parameters as follows: sensitivity = 90, shutter = 70, frame rate = 30, min brightness = 25, min area = 5, and max area = 1000.

### 4.4. CD73 Enzymatic Activity Measurements

The CD73/NT5E activity of the iodinated MSC exosomes was measured using the PiColorLock Gold Phosphate Detection System (Innova Biosciences, Cambridge, UK) according to the manufacturer’s instructions. In short, the buffer solution of exosome samples was first removed using an AcroPrep Advance 96-Well 100K MWCO filter plate (Pall, #8036) followed by 3 washes with 20 mM Tris HCl pH 7.4. Exosome samples were then incubated with 800 µM AMP solution (in 20 mM Tris HCl pH 7.4 buffer) for 1 h at 37 °C with 300 rpm in a microplate mixer. After that, the phosphate ions released from the hydrolysis of AMP by CD73 on the exosome were measured using the PiColorLock Gold Phosphate Detection System (Innova Biosciences, Cambridge, UK) according to the manufacturer’s instruction.

### 4.5. Statistical Tests

Statistical analyses were performed using Student’s *t* test. Data are shown as mean ± SD values. *p* values < 0.05 were considered statistically significant.

## 5. Conclusions

In this first study on the effects of labeling on the functional integrity of exosomes, we observed that the radioiodination of MSC exosomes, employing either the chloramine-T or Pierce iodination methods, compromised their functional integrity. This compromise was assessed through the measurement of CD73 enzymatic activity on the exosomes. However, it remains unclear whether other labeling techniques would yield similar effects on the functional integrity of exosomes. Nevertheless, this observation serves as a cautionary lesson, highlighting the importance of assessing functional integrity when labeling exosomes for biodistribution studies. The loss or compromise of functional integrity, synonymous with structural denaturation, may make the labeled exosomes susceptible to physiological degradation when administered in vivo. As a result, the biodistribution of such labeled but denatured exosomes will not reflect that of functionally intact exosomes. 

## Figures and Tables

**Figure 1 ijms-25-03742-f001:**
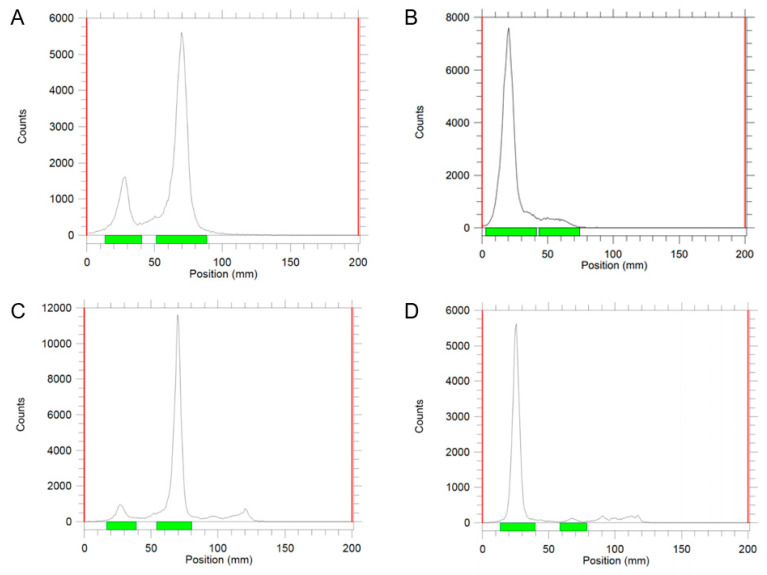
iTLC chromatograms of ^131^I-radiolabeled exosomes by chloramine-T, before (**A**) and after purification (**B**). iTLC chromatograms of ^131^I-radiolabeled exosomes by Pierce iodination reagent, before (**C**) and after purification (**D**). Green labels represent the development distances of species for calculating area under the curve.

**Figure 2 ijms-25-03742-f002:**
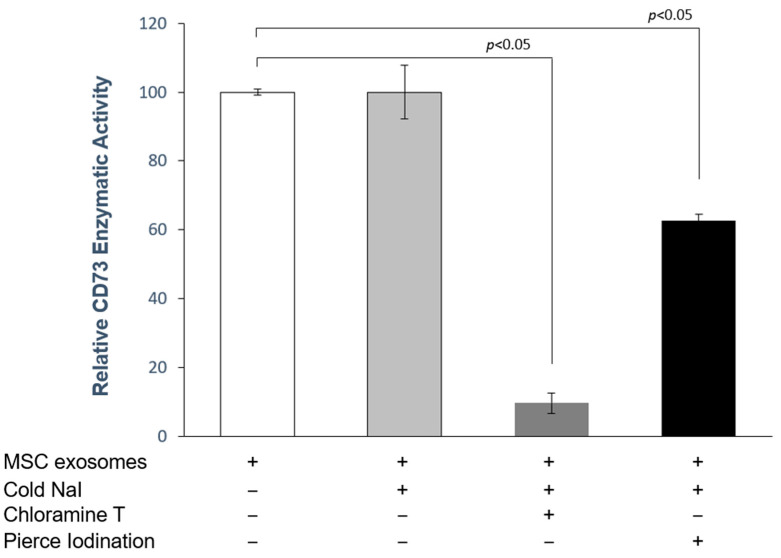
CD73 enzymatic activities of I-labeled exosomes through chloramine T and Pierce iodination. The enzymatic activities were normalized to the activity of the exosome input. Data are shown as mean ± SD values, n = 3.

**Figure 3 ijms-25-03742-f003:**
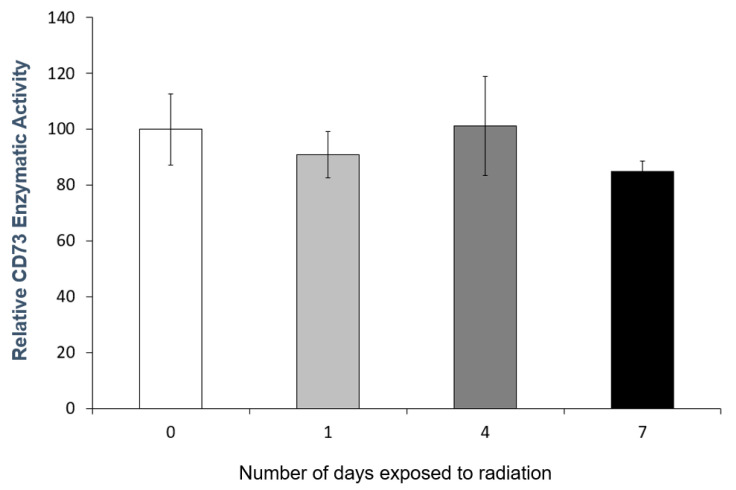
CD73 enzymatic activities of exosomes at different time points after exposure to ~12 mCi which is 4× radiation strength used in the labeling of exosomes. Data are shown as mean ± SD values, *n* = 3. There was no statistically significant difference in CD73 enzymatic activity at all time points compared with no treatment.

**Table 1 ijms-25-03742-t001:** Iodine-labeled exosomes with CD73 activity and particle size after labeling. Data are shown as mean ± SD values, *n* = 3.

	CD73 Activity	Particle Size
No labeling	100%	131.25 ± 15.91 nm
Chloramine T	9.7 ± 3.1%	148.39 ± 11.85 nm
Pierce	66.6 ± 1.9%	132.57 ± 27.9 nm

## Data Availability

Data contained within the article.

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
