# Peer review of "Standard Radio-Iodine Labeling Protocols Impaired the Functional Integrity of Mesenchymal Stem/Stromal Cell Exosomes"

_ijms, 2024, doi:10.3390/ijms25073742_

Round 1
Reviewer 1 Report
Comments and Suggestions for Authors
Please see attached document

Moderate editing of English language required
Author Response
- Please add a paragraph on the different and main sources, trilineage differentiation ability, accepted phenotypes and ISCT surface markers in the introduction.
We have added the below paragraph to the introduction section, line 37
MSCs are minimally defined by International Society for Cellular Therapy (ISCT) as being plastic-adherent when maintained in standard culture conditions, express CD105,CD73 and CD90, and lack expression of CD45, CD34, CD14 or CD11b, CD79a or CD19 and HLA-DR surface molecules and MSC must differentiate to osteoblasts, adipocytes and chondroblasts in vitro [1]. They could be isolated from diverse sources including bone marrow, adipose tissue, and umbilical cord blood [2].
References
- Galipeau, J. and L. Sensebe, Mesenchymal Stromal Cells: Clinical Challenges and Therapeutic Opportunities. Cell Stem Cell, 2018. 22(6): p. 824-833.
- Bhattacharya, P., et al., Efferocytes release extracellular vesicles to resolve inflammation and tissue injury via prosaposin-GPR37 signaling. Cell Rep, 2023. 42(7): p. 112808.
- While exosomes are one of the mechanisms by which MSC function and that is the focus of your current study; please acknowledge there are several other mechanisms by which MSC function and are used towards therapies. Please include this in the introduction before youbegin discussing your subjects.
Several other mechanisms have been proposed, with one of the more popular ones being efferocytosis [1]. However, it is now suggested that this mechanism may also involve extracellular vesicles (EVs) [2]. A comprehensive discussion of these other mechanisms would extend beyond the scope of this manuscript. Instead, we have modified two sentences to underscore the robustness of the data on EVs as the primary therapeutic mediator of MSCs.
- Line 55, we state, "The widely recognized therapeutic activity of MSCs against various diseases is now mainly attributed to small EVs or sEVs in the 50-200 nm size range [13-17]."
- Line 57, we mention, "In direct head-to-head comparisons between the therapeutic activity of MSCs and their EV products, MSC EVs could recapitulate the therapeutic effects of MSCs [7, 10, 18].
- Please indicate ‘n’ number for biological/technical repeats in all your figures legends with bar graphs. Please also indicate in the figure legends what the error bars indicate – mean? Median? Even if you have indicated them in material and methods.
Thank you for your comment, the suggested information is added now, line 132, line 141 and line 151. Data are shown as mean ± SD values, n=3.
- Please provide more details about the nanoparticle preparation and tracking.
To provide more details, we added the following
- Line 90: We investigated the effects of iodination using both chloramine-T and Pierce Iodination methods on the functional integrity of MSC exosomes produced by a monoclonal immortalized E1 mesenchymal stem cell line as described in the method by monitoring the enzymatic activity of CD73, a surface antigen on MSC exosomes. The use of monoclonal immortalized MSCs reduces variability in MSC exosomes caused by differences in EV-producing cells.
- Line 208: The exosome used in the manuscript was generated from a single MSC clone which is a monoclonal immortalized E1-MYC 16.3 human ESC-derived mesenchymal stem cell line [33] which Immortalized E1-MYC 16.3 human ESC-derived mesenchymal stem cells were cultured in DMEM with 10% fetal calf serum.
- Line 226: The MSC−exosomes preparations were characterised as per MSC-sEV defining identity proposed by a consensus of ISCT, ISEV, ISBT and SOCRATES [13]. Protein concentration was assayed using a Coomassie Plus (Bradford) Assay Kit (ThermoFisher Scientific, Waltham, MA, USA) and cholesterol concentration was assayed using Cholesterol Quantitation Kit (Abcam, Cambridge, UK). The expression of CD81 and CD73 in the preparations confirmed by western or ELISA [8, 35, 36].
- The nanoparticle tracking analysis is described in 4.3. Nanoparticle Tracking Analysis
- Are you able to provide any images of NP tracking?
We have the nanoparticle tracking images but due to the page limitation (it is a communication), we don’t think these are essential information.
- Do you have any evidence indicating that your radio labelling will not be cytotoxic?
We do not have evidence regarding the cytotoxicity of our radio labeling since we haven't conducted any labeling on living cells. Speculating on the cytotoxicity of exosomes iodinated with radioactive iodine is premature, as we haven't successfully labeled our exosomes while maintaining their integrity. Although numerous papers from the 1970s and 1980s discuss the iodination of cell membranes, there is no information available regarding the extent of cytotoxicity observed in these experiments.
- While your work is promising – more work is needed for this to reach publication quality.
We do not intend this to be a full-length manuscript but rather as a short communication to alert our fellow researchers on the pitfalls of iodinating exosomes with radioactive iodine using the current widely used iodination protocols.
Reviewer 2 Report
Comments and Suggestions for Authors
The manuscript entitled “Standard Radio-Iodine Labeling Protocols Impaired the Func-2 tional Integrity of MSC Exosomes” by Chang-Tong Yang et al. represents the study of iodine labeling on the integrity of small extracellular vesicles (exosomes) by determining the enzymatic activity of CD73 surface receptor. This is very important study to address the current challenges of exosome bio distribution after their administration in vivo. The manuscript would be interesting for the reader of the journal. However, some points have to be clarified before publication.
1) In my view additional characterization of exosomes is demanded. For example, TEM\SEM, and especially western blot to provide information about membrane receptors of exosomes.
2) I would suggest move Figure 3 and appropriate discussion to the main text.
3) Figure 2. It is unclear which type of labeling was used in time lapse experiments. Please provide an additional description.
Author Response
The manuscript entitled “Standard Radio-Iodine Labeling Protocols Impaired the Func-2 tional Integrity of MSC Exosomes” by Chang-Tong Yang et al. represents the study of iodine labeling on the integrity of small extracellular vesicles (exosomes) by determining the enzymatic activity of CD73 surface receptor. This is very important study to address the current challenges of exosome bio distribution after their administration in vivo. The manuscript would be interesting for the reader of the journal. However, some points have to be clarified before publication.
1) In my view additional characterization of exosomes is demanded. For example, TEM\SEM, and especially western blot to provide information about membrane receptors of exosomes.
As per our response to reviewer 1’s question 4, we used a monoclonal immortalized line i.e. the same cell for more than 100 preparations for the last ten years. The characterization of exosomes is in accordance with the ISCT, ISEV, ISBT and SOCRATES recommendations.
2) I would suggest move Figure 3 and appropriate discussion to the main text.
A good point. Done, see revised manuscript, line 109
3) Figure 2. It is unclear which type of labeling was used in time lapse experiments. Please provide an additional description.
In this time lapse experiments, we did not label the exosomes. We exposed the exosomes to radiation that has a radiation strength equivalent to 4 times that used in labelling, line 146
Reviewer 3 Report
Comments and Suggestions for Authors
In this manuscript, the authors investigated the effects of radioactive iodine on exosomes from mesenchymal stem/stromal cells (MSCs). The results showed that standard radio-iodine labeling protocols impaired the functional integrity of MSC exosomes through abrogating CD73 enzymatic activity. Overall, this is an interesting study and my specific comments are listed below.
1. The characterization of MSC exosomes should be shown. For example, the authors can show us the size distribution of exosomes by NTA. The exosome markers can be shown by flow cytometry or western blots.
2. In Figure 1 and 2, the statistical analysis results should be shown in the figures.
3. Please investigate more MSC exosomes markers besides CD73. I think it is hard to make an accurate conclusion by just using one marker.
4. It’s better to show the physiological effect of radioactive iodine on MSC exosomes, at least the cell-based assay is required and the animal model is encouraging.
Comments on the Quality of English LanguageNo
Author Response
In this manuscript, the authors investigated the effects of radioactive iodine on exosomes from mesenchymal stem/stromal cells (MSCs). The results showed that standard radio-iodine labeling protocols impaired the functional integrity of MSC exosomes through abrogating CD73 enzymatic activity. Overall, this is an interesting study and my specific comments are listed below.
- The characterization of MSC exosomes should be shown. For example, the authors can show us the size distribution of exosomes by NTA. The exosome markers can be shown by flow cytometry or western blots.
The exosome used in the manuscript was generated from Immortalized E1-MYC 16.3 human ESC-derived mesenchymal stem cell line. Its characterisation as per MSC-sEV defining identity proposed by a consensus of ISCT, ISEV, ISBT and SOCRATES has been described several times in previous reports. The characterisation by flow and western has also been described. Please refer to our response to reviewer 1 as followings.
1) Line 90: We investigated the effects of iodination using both chloramine-T and Pierce Iodination methods on the functional integrity of MSC exosomes produced by a monoclonal immortalized E1 mesenchymal stem cell line as described in the method by monitoring the enzymatic activity of CD73, a surface antigen on MSC exosomes. The use of monoclonal immortalized MSCs reduces variability in MSC exosomes caused by differences in EV-producing cells.
2) Line 208: The exosome used in the manuscript was generated from a single MSC clone which is a monoclonal immortalized E1-MYC 16.3 human ESC-derived mesenchymal stem cell line [33] which Immortalized E1-MYC 16.3 human ESC-derived mesenchymal stem cells were cultured in DMEM with 10% fetal calf serum.
3) Line 226: The MSC−exosomes preparations were characterised as per MSC-sEV defining identity proposed by a consensus of ISCT, ISEV, ISBT and SOCRATES [13]. Protein concentration was assayed using a Coomassie Plus (Bradford) Assay Kit (ThermoFisher Scientific, Waltham, MA, USA) and cholesterol concentration was assayed using Cholesterol Quantitation Kit (Abcam, Cambridge, UK). The expression of CD81 and CD73 in the preparations confirmed by western or ELISA [8, 35, 36].
4) The nanoparticle tracking analysis is described in 4.3. Nanoparticle Tracking Analysis
- In Figure 1 and 2, the statistical analysis results should be shown in the figures.
The p-value is now added, line 142, 153
- Please investigate more MSC exosomes markers besides CD73. I think it is hard to make an accurate conclusion by just using one marker.
We are uncertain which conclusion the reviewer is addressing. If the reviewer is referring to our assertion that our preparation constitutes an MSC exosome preparation, we acknowledge that relying solely on CD73 may not be accurate for such a conclusion. We direct the reviewer to our response to their first question for clarification. However, if the reviewer is addressing our conclusion regarding the compromised functional integrity of the exosomes due to the loss of CD73 activity, we respectfully disagree. This is because the reduction in CD73 activity signifies structural alterations, and CD73 serves as a crucial defining characteristic of MSC exosomes.
- It’s better to show the physiological effect of radioactive iodine on MSC exosomes, at least the cell-based assay is required and the animal model is encouraging.
We are uncertain about the reviewer's use of the term 'physiological,' as it typically pertains to living entities, whereas MSC exosomes are non-living entities. Considering that MSC exosomes are biochemical entities secreted by living cells, we conducted experiments to assess the impact of radioactive iodine on one of their biochemical functions, specifically CD73 activity. Given that iodination disrupts CD73 enzymatic activity, the functional integrity of MSC exosomes is compromised. Consequently, any effects observed from iodinated MSC exosomes on cells or animal models cannot be attributed to the MSC exosomes themselves, but rather to the compromised functionality of the exosomes.
Reviewer 4 Report
Comments and Suggestions for Authors
The manuscript titled: Standard Radio-iodine Labeling Impaired the Functional Integrity of MSC Exosomes, is well written, structured and organized. It presents novelty, however before publication some details need to be clarified:
What advantages do MSC exosomes have over MSCs as therapeutic agents?
How does the prevailing understanding suggest that MSC exosomes mediate their therapeutic impact?
What is the primary obstacle to the clinical translation of MSC exosomes?
Why is radioactive iodine considered a potential candidate for addressing the issue of exosome labeling?
What effects do iodination methods have on the structural or functional activities of MSC exosomes?
How might the biodistribution of iodinated exosomes differ from that of functionally intact ones?
Author Response
The manuscript titled: Standard Radio-iodine Labeling Impaired the Functional Integrity of MSC Exosomes, is well written, structured and organized. It presents novelty, however before publication some details need to be clarified:
What advantages do MSC exosomes have over MSCs as therapeutic agents?
We have added the following sentence in the introduction section, line 61
Compared with MSC cell-based therapy, exosome-based therapy offers tremendous advantages over cell- based therapy. It is smaller in size and non-living; therefore it is safer, easier to store, transport, and administer (see review [20]).
How does the prevailing understanding suggest that MSC exosomes mediate their therapeutic impact?
This is discussed in the introduction section with references, line 43-53
What is the primary obstacle to the clinical translation of MSC exosomes?
Like we mentioned in the abstract, the clinical translation of MSC exosomes is hindered by an incomplete understanding of their biodistribution after administration. And this represents one of the primary obstacles.
Why is radioactive iodine considered a potential candidate for addressing the issue of exosome labeling?
We have added the following sentence in the discussion section line 163.
Here we consider radiolabelling exosomes with iodine is because (1) iodine labeling is a well-established method; (2) its isotopes 131iodine has relatively long half-life (8 days), suitable for long-tracking kinetics, and its 125iodine labeled exosomes for PET imaging; (3) iodine can be labeled to the proteins in exosomes covalently or directly to membranes without genetic manipulation of the exosome-producing cells.
How might the biodistribution of iodinated exosomes differ from that of functionally intact ones?
The loss of CD73 enzymatic activity is symptomatic of protein denaturation. One expectation would that exosomes with denatured membrane proteins are likely to be cleared rapidly by the reticuloendothelial system (RES)
Round 2
Reviewer 2 Report
Comments and Suggestions for Authors
Authors revised their manuscript accordingly. The manuscipt can be accepted for publication.
Reviewer 3 Report
Comments and Suggestions for Authors
The authors revised their manuscript and provided relatively reasonable explanations for my questions. Some details were also added into this manuscript. The data in this study are integrated and more convincing. I have no more questions for the current version.
Reviewer 4 Report
Comments and Suggestions for Authors
Dear Authors,
I recommend the publication of the manuscript, taking in account that all the the comments of the reviewer were answered.
The manuscript presents novelty and relevance in the field of the journal.